# Angiogenic Properties of Placenta-Derived Extracellular Vesicles in Normal Pregnancy and in Preeclampsia

**DOI:** 10.3390/ijms22105402

**Published:** 2021-05-20

**Authors:** Natalia Gebara, Yolanda Correia, Keqing Wang, Benedetta Bussolati

**Affiliations:** 1Department of Molecular Biotechnology and Health Sciences, University of Torino, 10124 Torino, Italy; Natalia.gebara@unito.it; 2Aston Medical Research Institute, Aston Medical School, Aston University, Birmingham B4 7ET, UK; correia.yolanda94@gmail.com (Y.C.); k.wang@aston.ac.uk (K.W.)

**Keywords:** extracellular vesicles, preeclampsia, angiogenesis

## Abstract

Angiogenesis is one of the main processes that coordinate the biological events leading to a successful pregnancy, and its imbalance characterizes several pregnancy-related diseases, including preeclampsia. Intracellular interactions via extracellular vesicles (EVs) contribute to pregnancy’s physiology and pathophysiology, and to the fetal–maternal interaction. The present review outlines the implications of EV-mediated crosstalk in the angiogenic process in healthy pregnancy and its dysregulation in preeclampsia. In particular, the effect of EVs derived from gestational tissues in pro and anti-angiogenic processes in the physiological and pathological setting is described. Moreover, the application of EVs from placental stem cells in the clinical setting is reported.

## 1. Introduction

Extracellular vesicles (EVs) have been established as a means of cellular communication and are involved in physiological and pathophysiological processes through the transfer of bioactive molecules such as proteins, RNAs and lipids [1]. EVs are released by all cell types and can be found in all body fluids. The subtype and composition of EVs are dependent on the route of EV generation and parental origin, respectively [2]. The content of EVs can be packaged based on signals received from other cells or environmental factors such as oxygen, glucose concentration and sheath stress. In pregnant women, trophoblast-derived EVs are found in the blood [3], amniotic fluid [4], and urine [5]. These EVs have placenta-specific markers such as HLA-G [6], syncytin-1 [7] and placental-type alkaline phosphatase (PLAP-1) [8]. Due to their small size, EVs can cross the placental barrier, thereby enabling feto–maternal communication throughout the pregnancy [9] (Figure 1). Given that EVs are involved in a wide range of processes, it is no surprise that they play an important role in pregnancy. EVs regulate various normal physiological processes during pregnancy, including implantation of the embryo by regulation of the endometrium, trophoblast invasion, immune regulation of maternal responses and spiral artery remodeling [10]. Changes in the total number, content, and bioactivity of EVs have been reported in pregnancy complications such as preeclampsia (PE) [11].

Angiogenesis, the process by which blood vessels form for the delivery of nutrients to the body, is necessary both prior to and during pregnancy. A balance between pro and anti-angiogenic factors during pregnancy is key to a successful pregnancy. Imbalance of angiogenic factors, and subsequently, widespread endothelial dysfunction is considered the hallmark of pregnancy-related diseases such as preeclampsia (PE) [12]. The clinical manifestation of PE includes the development of hypertension after 20 weeks of gestation [13] and the coexistence of either proteinuria or other maternal organ dysfunction such as renal insufficiency, liver dysfunction, neurological features including headache or visual disturbances, hemolysis or thrombocytopenia, pulmonary edema [13,14,15]. Moreover, PE increases the risk of maternal and perinatal mortality and morbidity, and is associated with future cardiovascular disease risk [16]. Preeclampsia is classified as a new onset hypertensive pregnancy disorder. Depending on the time of delivery, early-onset preeclampsia manifests before the 34th week of gestation and is described as a fetal disorder that is associated with placental dysfunction and adverse fetal outcomes. Late-onset preeclampsia requires delivery at or after the 34th week of gestation and is considered as a maternal disorder associated with endothelial dysfunction and end organ damage [13].

Despite great scientific advances, PE is still defined as a “disease of theories”. Insufficient placenta development including abnormal spiral artery remodeling, placental hypoxia, oxidative stress, impaired angiogenesis and insufficient placental perfusion contribute to the development of preeclampsia [17,18,19]. Over the last two decades, increasing evidence suggests that angiogenic factors imbalance plays a pivotal role in PE pathogenesis [20,21]. Soluble fms-like tyrosine kinase 1 (sFlt-1), a soluble form of vascular endothelial growth factor receptor 1 (VEGFR-1) extracellular ligand-binding domain, is the key anti-angiogenic factor released by placenta during pregnancy. This soluble form of VEGFR-1 can bind to all isoforms of VEGF [22,23], as well as to the PlGF. Indeed, sFlt- 1 may act as a decoy receptor and hinder VEGF signaling through binding to its cognate receptors, thus inhibiting VEGF-mediated pro-angiogenic effects [24]. During normal pregnancy, the levels of sFlt-1 increase throughout gestation, providing a limiting and protective barrier for potentially unhealthy VEGF-dependent over signaling [25,26]. However, maternal circulating sFlt-1 levels are significantly increased in PE compared to normal pregnancy prior to the onset of the disease [27]. Furthermore, in vivo studies have showed that exogenous administration of sFlt-1 led to typical PE-like symptoms such as hypertension and glomerular endotheliosis in pregnant rats [20], whereas reduced circulating levels of free sFlt-1 below critical threshold rescued the damaging effects of sFlt-1 [21,28,29,30]. Another anti-angiogenic factor released by placenta is soluble endoglin (sEng), a soluble form of the coreceptor of transforming growth factor-β (TGF-β), which acts as a decoy receptor of TGF-β leading to a decrease in angiogenesis [31]. Not surprisingly, harmful molecules released from the placenta may reach the fetal circulation, causing endothelial dysfunction in the fetus. Indeed, many reports [32,33] have described fetoplacental endothelial dysfunction associated with preeclamptic pregnancies (Figure 1). A vicious cycle is therefore created, which is to the detriment of the mother and the fetus.

In pregnancy, EVs can stimulate or impede angiogenesis according to their content and molecular expression, which in turn, is dependent on the cell of origin and on its pathophysiological state. For instance, hypoxic conditions observed in the first trimester of the pregnancy, modulate the release of EVs from cytotrophoblast and placental cells, both in number and as cargo [34]. As the angiogenic aspects of EVs in pregnancy-related diseases are relatively new, this review will present the current studies related to EVs and their role in pregnancy and its associated complications.

## 2. Pro-Angiogenic Functions of EVs in Healthy Pregnancy

The influence of EVs on the angiogenic process has been reported in various in vitro and in vivo studies [12]. EVs are considered to modulate the angiogenic process through the transfer of several molecules including small RNA species and proteins. In particular, small RNA species, such as microRNA, are transferred by EVs and are considered pivotal for the reprogramming of target cells [35]. Lombardo et al. reported that miRNA-126 and p-STAT5 in endothelial-derived EVs are responsible for IL-3-mediated paracrine pro-angiogenic signals [36]. Similarly, endothelial-derived exosomes require miRNA-214 to simulate angiogenesis in recipient cells [37]. In vivo, injection of EPC-derived EVs containing the angiogenic miRNAs miRNA-126 and miRNA-296 in a mouse hind limb ligation model significantly increased capillary density and blood perfusion [38]. Several other microRNAs present in the EV cargo have been reported to be involved in the induction of angiogenesis, including miRNA-125a [39], miR31 [40] and miRNA-150 [41]. In parallel, several proteins released by EVs have also been found to be involved in the modulation of angiogenesis, including VEGF [42,43], FGF-2 [44], PDGF [43,45,46], c-kit [47], sphingosine-1-phosphate [48], regulated on activation of normal T cell expression and secretion [48], CD40L [49], C-reactive protein [50,51,52], metalloproteases [49,53], stem cell factor [54], and urokinase-type plasminogen activator. Furthermore, EVs can promote angiogenesis through the transfer of key lipids and proteins involved in activation of PI3K [55], extracellular signal-regulated kinase 1 and 2 [56,57,58], Wnt4/βcatenin [59], and nuclear factor-κB [60,61] pathways.

In pregnancy, a large number of studies focus on the activity of EVs released by maternal and fetal cells, including endothelial cells, immune cells, trophoblast and stem cells with the latter being present in the umbilical cord, placenta, amniotic fluid and amniotic membranes (Figure 1). These EVs are capable of inducing tissue regeneration and angiogenesis and can be found in maternal circulation starting from 7 weeks of gestation [62], in amniotic fluid [4], and urine [5]. The heterogeneity of EVs produced in this environment results in a plethora of pro-angiogenic factors being delivered to support the physiological development of normal pregnancy and the maintenance of endothelial homeostasis (Table 1). The following paragraphs will discuss the role of healthy pregnancy-related EVs in angiogenesis, according to their different tissue types of origin.

### 2.1. Maternal Blood EVs during Pregnancy

The concentration and profile of placental EVs in normal pregnancies has been extensively studied. Salomon et al., demonstrated that the concentration of EVs in maternal plasma increased more than two-fold towards the end of gestation [3]. The EVs in maternal serum showed different functional properties at different gestation. The ability of EVs to modulate endothelial cell migration decreased significantly over time, implying that EVs are less bioactive as the pregnancy progresses. As the EVs profile and functional properties may significantly change throughout gestation, it is important to study the different EVs populations to fully understand their function. Moreover, while EVs prepared from either maternal circulation or cord blood can enhance endothelial cell proliferation, migration, and promoted tube formation, the activity of EVs isolated from maternal serum is higher than that of EVs from cord blood. This is likely due to the differential expressions of microRNAs involved in regulating cell migration in maternal EVs and cord EVs, including miRNA-210-3p, miR376c-3p, miRNA-151a-5p, miRNA-296-5p, miRNA-122-5p, and miRNA-550a-5p [63].

### 2.2. Human Umbilical Cord Mesenchymal Stem Cells (UC-MSCs)-Derived EVs

UC-MSCs are a well-known source of EVs with pro-angiogenic activity. They are involved in stimulating endothelial cell migration, proliferation, and angiogenesis in vitro and activating the Wnt/β-catenin pathway [59]. Xiong et al. [64] characterized the angiogenic content of UC-MSC-EVs, which were produced under hypoxic conditions, in vitro [64]. They found that large number of pro-angiogenic factors, including VEGF, VEGFR-2, MPC-1, angiogenin, tie-2/TEK and IGF were present in these EVs. Moreover, UC-MSC-derived EVs from a healthy pregnancy in piglets showed upregulation of miRNA-150, which was directly correlated with higher expression of VEGF and Notch1, underlining exosome miRNA-150 as a major regulator of angiogenesis in utero [45].

### 2.3. Placental Explant EVs/Trophoblasts

EVs derived from placental tissue have been singled out for their tissue repair properties via angiogenesis regulation. The interest in placental derived-EVs offers us additional information on the possible pathway those EVs can regulate during pregnancy. Salomon et al. extracted EVs from placental MSCs and subjected them to in vitro and in vivo functional assays [34]. The results showed an increase in endothelial cell migration and vascular tube formation. In addition, Salomon et al. discovered multiple miRNAs enclosed in placental EVs, such as miR486-1-5p and miR486-2-5p, which target the pathways responsible for migration, placental development, and angiogenesis [34]. Moreover, under hypoxic conditions, the release of exosomes and the expression profile of pro-angiogenic factors were increased, indicating that changes in oxygen tension during early pregnancy may promote angiogenesis via release of placental MSCs-derived EVs [34]. EVs isolated from trophoblasts were found to release EMMPRIN, which is a protein that contributes to spiral artery remodeling [65]. EMMPRIN has been shown to promote both angiogenesis and metalloproteinase gene expression in endothelial cells of the umbilical cord and to upregulate VEGF production though HIF1α expressing fibroblasts [65]. In addition, EVs from trophoblast cells obtained via perfusion of placental explants were shown to contain endothelial Nitric oxide synthase (eNOS) [66]. Indeed, EVs could transfer eNOS to endothelial cells, resulting in vasodilation of the endothelium during normal pregnancies [65].

### 2.4. Amniotic Fluid Stem Cells EVs

Amniotic fluid stem cells (AFSC) have recently been of interest in regenerative medicine. AFSCs express high levels of stem cell receptor factor, c-kit (CD117) and display the characteristics of multipotent cells, and they can differentiate into all the germ cell layers because of their fetal but non-embryonic origin [67]. The paracrine effect of these cells has been attributed to signaling via EVs. AFSC-EVs contribute to multiple angiogenic processes that aid in gestational development [68]. A substantial amount of evidence demonstrates the pro-angiogenic, anti-apoptotic, and immunosuppressive effects of AFSC-EVs, both in vitro and in vivo models of organ damage [69,70]. Sedrakyan et al. [71] used the high expression of VEGFR1/2 on AFSC-EVs as an antagonist of VEGF-induced endothelial damage in a murine model of Alport syndrome [71]. The excess of VEGF was bound to the VEGFR receptors on the EVs, easing the symptoms of the disease.

### 2.5. Amnion EVs

The placenta is a complex organ responsible for most of the maternal and fetal health, and as such is composed of both maternal and fetal tissue. The membrane of this fetal tissue (amnion membrane) is comprised of cuboidal epithelial cells that stick to the basement membrane, and of an extracellular matrix with fibroblast-like cells. The 1 amnion epithelial cells released by the amnion membrane are represented by epithelial characteristics and membrane antigens. Administration of EVs released from these amnion epithelial cells have been used as a therapy in different diseases such as idiopathic pulmonary fibrosis [72]. However, their potential for therapy in pregnancy-related diseases has been poorly explored. More research is needed to conclusively understand and explore the potential of amnion EVs for therapy in pregnancy-related diseases [72].

## 3. Anti-Angiogenic Actions of EVs in Preeclampsia

As discussed above, placental-derived EVs play an important physiological role in mother and fetus communication through the delivery of a range of proteins, lipids, and nucleic acids [74]. On the other hand, alterations in the concentration and content of placenta-derived EVs are associated with pregnancy-related diseases, such as PE [75,76,77,78,79]. Angiogenic imbalance is considered to be the main contributor of endothelial dysfunction, in PE [80]. Flt-1 and Eng have been detected on the surface of placental-derived EVs [3,75,81]. These Flt-1 and Eng expressing EVs are capable of binding to pro-angiogenic factors, VEGF, PlGF and TGF-β [75]. Interestingly, the capacity of Eng expressing placental-EVs to bind TGF-β was much higher in comparison to the affinity for VEGF and PlGF to Flt-1 expressing EVs [76].

The anti-angiogenic effect of circulating EVs in PE appears to be, at least in part, mediated by the presence of Flt-1 or Eng in the form of membrane receptors on the EV surface, as these can act as decoy receptors by sequestering angiogenic soluble factors, similarly to the presence of the soluble forms. Tannetta et al. [82] showed that EVs isolated from preeclamptic placenta expressed higher level of Flt-1 in comparison to EVs isolated from normal placenta. Application of EVs containing sFlt-1 to HUVEC cells led to a reduction in tube formation, cell invasion and an increase in cellular permeability [82]. EVs isolated from first trimester placental explants treated with sera from either normal pregnant women or preeclamptic women showed no significant differences in their size or concentration. However, EVs isolated from preeclamptic sera-treated placental explants were able to cause endothelial cell activation due to up-regulation of High Mobility Group Box 1 (HMGB1) in EVs, a potent danger-associated molecular pattern/danger signal that can lead to sterile inflammation [83]. Furthermore, administration of platelet or endothelial cell-derived EVs to pregnant mice induced PE-like symptoms and decreased embryonic vascularization [84]. Han et al. showed that infusion of EVs derived from injured placenta rather than normal placenta led to the development of hypertension and proteinuria in pregnant mice. These EVs released from injured placenta caused disruption of endothelial integrity and enhanced vasoconstriction [85]. These studies highlighted a possible pathogenic role of circulating EVs in PE. Studies in vivo have also confirmed the role of EVs in affecting angiogenesis and inducing PE-like syndromes by administering EVs derived from pre-eclamptic placentas in mice. The injection of those EVs resulted in damage to the vasculature and poor fetal nutrition [81].

The shedding of EVs into the maternal blood in PE has also been shown to correlate with systolic blood pressure [86]. Neprilysin (NEP), a membrane-bound metalloproteinase that is involved in vasodilator degradation, has been directly correlated with hypertension (REF). Interestingly, NEP has been found on the surface of trophoblast-EVs. Moreover, Gill et al. [87] showed that the level of NEP in the EVs derived from placenta and syncytiotrophoblast cells were both augmented in preeclamptic women compared to normal pregnant women, indicating that increased levels of NEP-expressing EVs in the maternal circulation may play a role in causing PE-related symptoms including hypertension, heart failure [87]. Furthermore, placental trophoblast-EVs in PE contain reduced levels of endothelial nitric oxide synthase (eNOS), and thus reduced production of nitric oxide production, a potent vasodilator (59), suggesting that PE-associated EVs may contribute to decreased NO bioavailability, resulting in endothelial dysfunction. Additionally, EVs may also inhibit angiogenesis by low-density lipoprotein receptor-mediated endocytosis [88], and by CD36-dependent uptake of EVs and induction of oxidative stress [89,90].

Not surprisingly, the majority of studies that have investigated EV in the development of PE have focused on the effect of placental EVs on the endothelium. Vascular smooth muscle cells residing in vessels are also an important part of the processes that lead to placentation and vascular remodeling. EVs from extravillous trophoblasts were reported to promote vascular smooth muscle cell migration, thus suggesting their role in the remodeling of spiral arteries [91]. However, the possible contribution of EVs on vascular smooth muscle cells in the pathophysiology of preeclampsia have not yet been researched.

Beside membrane-bound proteins, EVs derived from gestational tissues during complicated pregnancies contain multiple miRNAs, which may play an anti-angiogenic role under these conditions, particularly in PE [33]. Cronqvist et al. [92] demonstrated that syncytiotrophoblast-derived EVs directly transferred functional placental miRNA that directly targeted Flt-1 mRNA to primary human endothelial cells and subsequently regulated the expression of Flt-1, suggesting that miRNA enclosed in placental EVs may directly affect the maternal and fetal endothelial function through regulating angiogenic factors [92]. The miRNAs most commonly involved in angiogenic processes are outlined below (Table 2) [11]. For example, miRNA-210, which has an anti-angiogenesis property is actively secreted in placental-EVs. It has been found in the circulation of pregnant women and the level of miRNA-210 was elevated in women with PE and hypoxia [93]. In addition, EVs derived from trophoblast and endothelial cells contained miRNA-210 and its action may be mediated through directly targeting ERK [43]. miRNA-520c-3p, another miRNA found in trophoblast-EVs, can reduce invasiveness in cancerous cells [94,95]. Interestingly, Takahashi et al. [95] showed that MiRNA-520c-3p was present in first-trimester trophoblast cells and trophoblast cell lines and that overexpression of MiRNA-520c-3p in EVs significantly inhibited cell invasion by targeting CD44. The results showed that EVs produced by cells with miRNA-520c-3p overexpression significantly reduced invasiveness via repression of CD44. Moreover, several miRNAs which have been previously associated with hypertension have been found to be differentially expressed in preeclamptic EVs, including miRN26b-5p, miRNA-7-5p and miR181a-5p [96]. In addition, placental 19 miRNA cluster (C19MC), a cluster almost exclusively expressed on the placenta [97], has been reported to be altered in PE [94].

Altogether, the studies described above suggest that EVs derived from preeclamptic placentas play a direct role in endothelial dysfunction and contribute to PE development through their membrane-bound proteins and the release of internal molecules such as anti-angiogenic proteins and miRNAs. However, recent studies have challenged these theories. It has been shown that EVs isolated using the common methods may cause contamination of soluble bioactive factors that may affect their resultant bioactivities [98]. O’Brien et al. [99] showed that placental explant-derived EVs had no significant effect on angiogenesis in vitro when EVs were administered alone, in contrast with the effect of EVs administered with conditioned medium. These data may suggest that the endothelial dysfunction associated with PE is linked with the action of soluble factors rather than EVs. To fully elucidate this point, future research on EVs bioactivity requires more stringent design and planning. Moreover, EVs from different sources must be researched on different platforms (in vitro, in vivo, organ on a chip) and on different functional assays (invasion, angiogenesis, apoptosis, etc.).

## 4. Potential Clinical Applications of Pregnancy Related EVs in Therapy and Diagnostic Applications

Stem cell from gestational tissues such as the umbilical cord, the placenta, fetal membranes (amnion) and amniotic fluid appear to be a rich source of EVs with promising therapeutic activity. The advantage of therapeutic use of EVs relies on the lack of replication and immunogenicity in respect to the parental cells. The administration process is more straightforward than the one for cells, as they can be stored and generated with a higher economic benefit. Furthermore, unlike stem cells, no cases of tumor-related downfalls have been reported so far [101]. Placental tissue-derived EVs have been studied in clinical trials since the 2000s [102]. Currently, most of these clinical trials of EVs-based therapies are focused on conditions such as melanoma [103], cancers [104], type II diabetes (NCT02138331) and cutaneous ulcers (NCT02565264), rather than pregnancy-related disorders. Recently, the FDA approved a clinical trial using amniotic fluid-sourced acellular components for COVID-19 treatment (NCT04384445). This treatment claims to contain naturally occurring hyaluronic acid, proteins, and exosomes, administered intravenously with 1 mL, containing 2–5 × 10^11^ particles. However, the characterization of the amniotic fluid vesicles was not provided [105]. Unfortunately, these studies only looked at small cohorts of people. Nevertheless, these studies showed that using EVs as therapy could be a viable option, although more studies are needed to conclusively understand the role of EVs and how EVs exert their actions under different conditions [12]. The proper characterization of cell-secretome other than EVs and the factors responsible for effects observed represent an emerging issue in EV therapy.

Based on their origin and content, EVs can also be applied for diagnostic purposes. The characteristics of EVs would make them the perfect candidate to act as a marker of genetic disorders or early signs of pregnancy-related complications. For instance, the levels of Syncytin-2, a protein specific to the placenta, has been shown to be significantly reduced in serum-derived EVs from preeclamptic patients and could represent a potential biomarker to predict the development and severity of preeclampsia [7]. Miranda et al. [10] previously used EV concentration and content to evaluate the stage of fetal development. Considering these EVs can be extracted and analyzed directly from a maternal blood sample, this presents an opportunity for the development of a non-invasive test in contrast to the amniocentesis. This opportunity for a non-invasive test should be explored as it would represent both a less invasive technique as well as a less risky method of analysis as the risk to the fetus would be greatly diminished.

## 5. Conclusions

EVs are important cell-derived bio-factors involved in cell-to-cell communication, and they are highly implicated in the physiological development of normal pregnancy and in the maintenance of endothelial homeostasis. Multiple sources of angiogenesis-related EVs are released from fetal tissues and are present in the maternal circulation. A better characterization of their differential cargoes during pregnancy appears to be an interesting field of research. Nevertheless, the pro-angiogenic and pro-regenerative EVs from placental tissues can be further exploited in different clinical settings. In particular, placental stem cell-derived EVs or amniotic fluid EVs are subject to increased interest for organ regeneration and repair. Finally, knowledge of EV function and cargo might be instrumental for the diagnosis and therapy of pregnancy-related disorders. Indeed, a large number of studies indicate that placenta-released EVs show altered phenotype and function. Placenta-derived EVs increase in maternal circulation in PE and when transferred in murine experiments, can highly affect endothelium in synergy with soluble factors to cause endothelial dysfunction.

However, knowledge of the involvement of EVs in placental angiogenic processes is still limited, and more studies are still needed, especially on the early processes occurring in the first trimester.

## Figures and Tables

**Figure 1 ijms-22-05402-f001:**
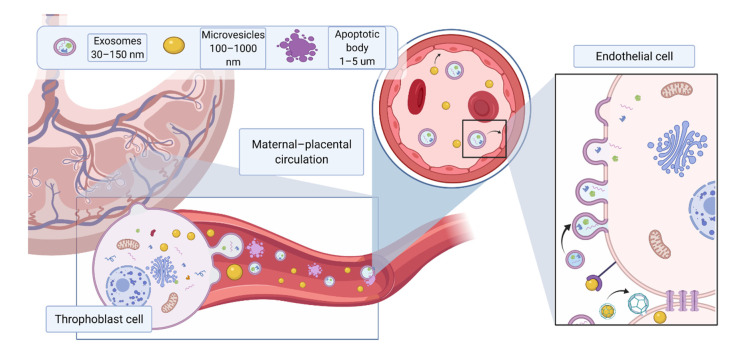
Different EV subtypes released from the placenta during pregnancy. The main subtypes of extracellular vesicles reported include exosomes (30–150 nm) formed from intracellular endosomal compartments and characterized by expression of tetraspanins (CD63, CD81, CD9); microvesicles (100 nm–1 μm), which originate from cell plasma membrane, characterized by CD 40; apoptotic bodies (>1 μm), which are released by apoptotic cells, express phosphatidylserine on their surface and caspase 3 and 7 internally. Due to their small size, EVs can cross the placental membrane and contribute to feto–maternal signaling. Placental EVs have been previously found in maternal circulation and shown to directly affect maternal endothelium. Created with BioRender.com (accessed on 18 May 2021).

**Table 1 ijms-22-05402-t001:** Angiogenic factors associated with EVs and their role in pregnancy.

Source	Bio-Factor/Functional Assay	Platform	Reference
Umbilical cord blood	miRNA-15, miRNA-150	In vitro	Luo et al., 2018 [45]
Umbilical cord	VEGF, VEGFR-2, MPC-1, angiogenin, tie-2/TEK and IGF	In vitro	(Xiong et al., 2018) [64]
Umbilical cord	Activation of Wnt/β-catenin pathway	In vitro	(Zhang et al., 2015) [59]
Placental explant EVs	Angiogenesis and migration	In vitro and in vivo	(Salomon et al., 2013) [34]
Trophoblast-EVs	EMMPRIN	In vitro and in vivo	(Balbi et al., 2019) [69]
Maternal–blood EVs	Shh	In vivo	(Martínez et al., 2006) [73]
Maternal–blood EVs	enhances endothelial cell proliferation, migration, and tube formation	In vitro	(Jia et al., 2018) [63]
Throphoblast-EVs	eNOS	In vitro	(Motta-Mejia et al., 2017) [66]
Placental explants	Flt/endoglin	In vitro	(D. Tannetta et al., 2014) [62]

**Table 2 ijms-22-05402-t002:** Angiogenic factors associated with EVs and their role in pregnancy.

Source of EVs	Bio-Factor/Functional Assay	Platform	Reference
Placental explants/trophoblast cell culture/maternal blood	s-Eng	in vitro and in vivo	(Chang et al., 2018) [100]
			(Schuster et al., 2020) [80]
			(D. S. Tannetta et al., 2013) [76]
			(Salomon et al., 2014) [3]
Placental explants/trophoblast cell culture/maternal blood	s-Flt	in vitro and in vivo	(Chang et al., 2018) [81]
			(Schuster et al., 2020) [80]
			(D. S. Tannetta et al., 2013) [75]
			(Salomon et al., 2014) [3]
Trophoblast EVs	NEP	In vitro	(Gill et al., 2019) [87]
Placental explants	VEGFR1/Endoglin	in vivo	(Tannetta et al. 2013) [82]
Placental explants	HMGB1	In vitro	(Xiao et al.,2017) [83]
Placental explant	miRNA-210	In vitro	(Anton et al., 2013) [93]
Placental explant	miRN26b-5p, miRNA-7-5p and miR181a-5p	In vitro	(Zhang et al., 2020) [96]
Placental explant	C19MC associated miRNA	In vitro	(Morelli et al., 2012) [94]
Maternal circulation	miRNA-486-1-5p, miRNA-486-2-5p	In vitro	(Salomon et al., 2013) [34]
Trophoblast EVs	miRNA-520c-3p	In vitro	(Takahashi et al., 2017) [95]
Serum derived EVs	Syncitin-2	X	(Vargas et al., 2014) [7]

## Data Availability

Not applicable.

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
