# Peer review of "Angiogenic Properties of Placenta-Derived Extracellular Vesicles in Normal Pregnancy and in Preeclampsia"

_ijms, 2021, doi:10.3390/ijms22105402_

Round 1

Reviewer 1 Report

The review paper "Angiogenic properties of placenta-derived extracellular vesicles in pregnancy and preeclampsia" by Gebara et al. outlines the implications of EV-mediated crosstalk in the angiogenic process in healthy pregnancy and its dysregulation in preeclampsia. 

A very elegant and interesting review. The topic is very important and relevant. congratulations for the authors.

There are only minor typos which needs to be assessed (starting by affiliations).

Author Response

To Reviewer 1

We thank the reviewer for the positive comments.

Minor concerns

English language and style are fine but minor spell checks are required.

We did our best to improve it.

Reviewer 2 Report

Thank you very much for allowing me to review this manuscript.
In my opinion the authors raise a very important topic and relevant issue. Unfortunately I cannot recommend this paper for publication in its current form.
-    The manuscript requires clarification. The data presented in this paper are given chaotically and it is not clear for the reader what new conclusions result from this paper.

A few specific concerns:

1 / the title of the manuscript should be more informative - is it about normal pregnancy and pregnancy complicated by preeclampsia or only in preeclamptic pregnancy?

2 / it would be worth giving a detailed description of the clinical presentation of preeclampsia  - it seems necessary to review the manuscript by a specialist obstetrician or perinatologist for better understanding of the clinical picture of preeclampsia 

3 / it would be good to provide the current guidelines and classification of preeclampsia

4 / references 74 and 76 are the same publication

5 / in line 40 on page 1 the authors refer to the work by Brown et al. 2018 but there is no such work in the list of publications - reference at the end of the manuscript

it would have to be tidied up and explained

6 / the final conclusions are too general and the question remains open, what new did the reader learn after reading this manuscript ?

7 / there are a bit of information and data are given chaotically - the authors jumping from one data / one information to another 

Author Response

To Reviewer 2

We thank the reviewer for the comments and the suggestions, that we followed in this new version of the manuscript.

The manuscript requires clarification. The data presented in this paper are given chaotically and it is not clear for the reader what new conclusions result from this paper.

We are sorry for the unclear writing and not providing a precise conclusion, the manuscript was extensively revised to address those issues.

The title of the manuscript should be more informative - is it about normal pregnancy and pregnancy complicated by preeclampsia or only in preeclamptic pregnancy?

The title of the review was changed to: Angiogenic properties of placenta-derived extracellular vesicles in normal pregnancy and in preeclampsia.

  1. It would be worth giving a detailed description of the clinical presentation of preeclampsia - it seems necessary to review the manuscript by a specialist obstetrician or perinatologist for better understanding of the clinical picture of preeclampsia 

The clinical presentation of preeclampsia was added: The clinical manifestation of PE includes the development of hypertension after 20 weeks of gestation[13] and the coexistence of either proteinuria or other maternal organ dys-function such as renal insufficiency, liver dysfunction, neurological features including headache or visual disturbances, hemolysis or thrombocytopenia, pulmonary edema [13–15]. Moreover, PE increases the risk of maternal and perinatal mortality and morbidity, and is associated with future cardiovascular disease risk [16]. See page 1 and 2, lines 44-50.

  1. It would be good to provide the current guidelines and classification of preeclampsia

We added the classification in early and late presentation.

Preeclampsia is classified as a new onset hypertensive pregnancy disorder. Depending on the time of delivery, early-onset preeclampsia manifests before the 34th week of gestation and is described as a foetal disorder that is associated with placental dysfunction and adverse foetal outcomes. Late-onset preeclampsia requires delivery at or after the 34th week of gestation and is considered as maternal disorder associated with endothelial dysfunction and end organ damage [13].  See page 2, lines 50-55.

  1. References 74 and 76 are the same publication

The referencing issue was fixed and ref. 76 deleted.

  1. In line 40 on page 1 the authors refer to the work by Brown et al. 2018 but there is no such work in the list of publications - reference at the end of the manuscript

The reference was added:

[13]. Brown, M.A.; Magee, L.A.; Kenny, L.C.; Karumanchi, S.A.; McCarthy, F.P.; Saito, S.; Hall, D.R.; Warren, C.E.; Adoyi, G.; Ishaku, S. Hypertensive disorders of pregnancy: ISSHP classification, diagnosis, and management recommendations for international practice. Hypertension 2018, 72, 24–43, doi:10.1161/HYPERTENSIONAHA.117.10803.

6.The final conclusions are too general, and the question remains open, what new did the reader learn after reading this manuscript?

We have changed the conclusion to be more informative:
EVs are important cell derived bio-factors involved in cell-to-cell communication, highly implicated in the physiological development of normal pregnancy and in the maintenance of endothelial homeostasis. Multiple sources of angiogenesis-related EVs are released from foetal tissues and are present in the maternal circulation. A better characterization of their differential cargoes during pregnancy appears as an interesting field of research. Nevertheless, the proangiogenic and proregenerative EVs from placental tissues can be further exploited in different clinical settings. In particular, placental stem cell-derived EVs or amniotic fluid EVs are getting increased interest for organ regeneration and repair. Finally, the knowledge on EV function and cargo might be instrumental for the diagnosis and therapy of pregnancy-related disorders. Indeed, a large number of studies indicate that placental released EVs show altered phenotype and function. In particular, placenta derived-EVs increase in maternal circulation in PE and when transferred in murine experiments can highly affect endothelium in synergy with soluble factors to cause endothelial dysfunction.

However, knowledge the involvement of EVs on placental angiogenic processes is still limited, and more studies are still needed, especially on early processes occurring in the first trimester.

  1. There are a bit of information and data are given chaotically - the authors jumping from one data / one information to another 

We did our best to ameliorate the data presentation. We moved or deleted text to clearly divide the presentation into a section on the angiogenic characteristics of EVs in healthy pregnancies, one on the altered characteristics of EVs in PE, and a third on the clinical relevance of EVs, including the exploitation of placental EVs and the diagnostic potential of EVs in PE.  We believe that the text flows better after these changes.

Round 2

Reviewer 2 Report

The manuscript has been improved